# Oral Administration of Probiotics (*Bacillus subtilis* and *Lactobacillus plantarum*) in Nile Tilapia (*Oreochromis niloticus*) Vaccinated and Challenged with *Streptococcus agalactiae*

Mateus Cardoso Guimarães [1,2], Isabel M. Cerezo [3], Miguel Frederico Fernandez-Alarcon [1],
Mariene Miyoko Natori [1], Luciana Yuri Sato [1,2], Camila A. T. Kato [1], Miguel Angel Moriñigo [3],
Silvana Tapia-Paniagua [3], Danielle de Carla Dias [1], Carlos Massatoshi Ishikawa [1], Maria José T. Ranzani-Paiva [1],
Luara Lucena Cassiano [4], Erna Elisabeth Bach [5], Patrícia B. Clissa [6], Daniele P. Orefice [6] and
Leonardo Tachibana [1,*]

1 Fisheries Institute/APTA/SAA, Aquaculture Research Centre, Av. Conselheiro Rodrigues Alves 1252, Vila Mariana, São Paulo 04014-900, Brazil
2 Fisheries Institute/APTA/SAA—Postgraduation Program, São Paulo 04014-900, Brazil
3 Department of Microbiology, Malaga University, Campus de Teatinos s/n, 29010 Málaga, Spain
4 Biological Institute—Postgraduation Program in Health, Food Security and Environmental in Agribusiness, Av. Conselheiro Rodrigues Alves 1252, São Paulo 04014-002, Brazil
5 Biological Institute, Av. Conselheiro Rodrigues Alves 1252, São Paulo 04014-002, Brazil
6 Butantan Institute, Av. Vital Brasil 1500, Butantã, São Paulo 05503-900, Brazil
* Correspondence: tachibana@pesca.sp.gov.br

**Abstract:** *Streptococcus agalactiae* is an important bacterial pathogen in intensive Nile tilapia production, causing high mortality rates and great economic losses. This work aimed to evaluate the Nile tilapia vaccination against *S. agalactiae* and fed with ration containing probiotic AQUA PHOTO® composed of *Bacillus subtilis* and *Lactobacillus plantarum*, on the immune response action and gut microbiota. The experimental design was completely randomized with five treatments (CON = control; ADJ = adjuvant; PRO = probiotic; VAC = vaccine; PRO + VAC = probiotic + vaccine) and five replicates. The vaccine (bacterin + adjuvant) was injected after 21 days (21d) of probiotic feeding and the vaccine was booster 14 days post-vaccination (35d). After 14 days of the booster (49d), the fish were challenged with *S. agalactiae* and observed for more than 14 days, completing 63 days. The immunized group showed a better survival rate (CON 40%; ADJ 57%; PRO 67%; VAC 87%; PRO + VAC 97%). The treatments VAC and PRO + VAC, after booster produced higher levels of IgM antibodies compared with the control from the same time. The combination of probiotic and vaccination provided better protection against *S. agalactiae* infection, directly affecting the gut microbiological profile. These results indicated the contribution of probiotic to the adaptive immune response through the modulation of the intestinal microbiota, improving the effect of the vaccination. In conclusion, AQUA PHOTO®, composed of *B. subtilis* and *L. plantarum*, orally administered to Nile tilapia vaccinated against and challenged with *S. agalactiae* increases protection from infection and modifies the intestinal microbiota profile of the host, promoting the microbiota balance and improving adaptive immune response.

**Keywords:** infection; immunization; gut microbiological profile; prophylaxis

## 1. Introduction

Modern aquaculture requires solutions to achieve improved growth performance and infectious disease prevention [1]. Current Brazilian fish culture uses intensive systems to maintain high results, which were over 841,005 tons in 2021 (4.7% increase compared to 2020), and 486,155 tons correspond to Nile tilapia (*Oreochromis niloticus*) production (63.5% of total fish production) [2].

*Streptococcus agalactiae*, also known as group B *Streptococcus* (GBS), is a Gram-positive pathogenic bacterial species, with occurrence in temperate and tropical regions, and it is a threat to animal and human health [3,4]. Streptococcosis is a fish disease which is transmitted horizontally by direct contact between infected and healthy animals or indirect contact through water and fomites. In the natural route of infection, the bacteria enter the intestinal tract by crossing physical and chemical barriers (serum lysozymes, mucus, among others). The infection can be systemically disseminated, affecting mainly the brain, kidney, and intestine, causing cell death [5–7]. The clinical signs of fish streptococcosis are lethargy, dorsal rigidity, and erratic swimming as a result of damage to the central nervous system, causing a chronic systemic inflammatory response [5].

Antibiotics are used to treat or prevent bacterial infections in animals, and its overuse may be a potential risk to the environment, food security and the spread of bacterial resistance [6,7]. The probiotics administration provided a notable positive effect on zootechnical performance parameters and can be an alternative strategy for disease control. This feed additive is defined as live microorganisms with a great capacity to improve host health, promoting a balance of intestinal microbiota [8]. Gram-positive and -negative bacteria and bacteriophages have been used as probiotics and have a beneficial impact on the host [9].

The ability of probiotics to improve host health is related to their mechanism of action that relate to competitive pathogens' exclusion, reduction of toxic amine production, apparent nutrient digestibility improvement and immune system stimulation. Thus, these benefits result in a better growth performance index [10]. According to [11], the oral administration of AQUA-PHOTO® (Biogenic group, São Paulo, Brazil), a commercial probiotic composed of *Bacillus subtilis* and *Lactobacillus plantarum*, (for seven days) modulated the intestinal microbiota profile and increased amylase activity, therefore improving the carbohydrates' diet utilization by Nile tilapia.

Vaccination is the best method to promote immunization against many infectious diseases, due to the memory cell activation and immune response against antigen [12]. In aquaculture, vaccines are administered via the oral, immersion, or intraperitoneal/intramuscular route [13,14]. The administration process must consider the pathogen characteristics, infection route, vaccine production technique (live or inactivated), immunologic memory and host immune system status, production system, handling, nutrition, cost and environmental conditions, among other factors [14–16].

Lazado et al. (2014) [16] showed that the vaccination effect in fish could be enhanced by feeding them probiotic diets, due to the interaction between the intestinal microbiota and the gut associated lymphoid tissue system (GALT).

This work aimed to evaluate the Nile tilapia vaccination against *S. agalactiae*, potentiated with probiotic AQUA PHOTO® on feeding, composed of *B. subtilis* and *L. plantarum*, on the immune response action and gut microbiota.

## 2. Materials and Methods

### 2.1. Inoculum of Streptococcus Agalactiae

The *S. agalactiae* serotype Ib (21171A) was obtained from PREVET Laboratory (Brazil) and specie determined by molecular testes (PCR) using a specific primer [17] and was cultivated in BHI broth medium for 48 h, 37 °C and diluted in NaCl 0.85% solution at $3.3 \times 10^4$ CFU mL$^{-1}$. The LD50 dose for the *S. agalactiae* strain (21171A) was previously determined [18].

### 2.2. Vaccine Preparation

The vaccine was developed by Biocamp Laboratory (Campinas, Brazil). The bacteria were cultivated in BHI broth up to $3.33 \times 10^8$ CFU mL$^{-1}$, inactivated with formalin and emulsified in adjuvant oil according to established protocols [19]. The solution was kept on ice until the moment of fish vaccination.

### 2.3. In Vivo Infection Assay

The experiments were conducted at the Aquaculture Research Centre/Fisheries Institute/APTA/SAA, São Paulo, Brazil. The experimental infection assay was carried out according to Resolution No. 592 of the Brazilian Federal Council of Veterinary Medicine and precepts of Ethical Principles of Animal Experimentation. This project was approved by the Fisheries Institute Ethics Committee (protocol No. 01/2018).

2.3.1. Control and Experimental Diets

The experimental diets were formulated according to the Nile tilapia nutritional requirements, described in [20] (Table 1).

**Table 1.** Formulation and estimated percentage composition of experimental and control diets of Nile tilapia (*O. niloticus*).

| Ingredient (g.100 g$^{-1}$) | Control Diet | Probiotic Diet |
| --- | --- | --- |
| Wheat meal | 14.18 | 14.18 |
| Corn | 21.79 | 21.79 |
| Poultry by-product meal | 14.11 | 14.11 |
| Bovine hemoglobin | 13.00 | 13.00 |
| Meat and bone meal | 12.00 | 12.00 |
| Soybean meal | 10.00 | 10.00 |
| Middling rice | 6.00 | 6.00 |
| Rice meal | 4.00 | 4.00 |
| Corn gluten meal | 1.66 | 1.66 |
| Soybean oil | 1.50 | 1.50 |
| Vitamin and mineral supplement (Premix) [1] | 0.50 | 0.50 |
| Probiotics (*L. plantarum* and *B. subtilis*) | 0.00 | 0.02 |
| Salt (NaCl) | 0.50 | 0.50 |
| DL-Methionine | 0.20 | 0.20 |
| Vitamin C (35%) | 0.20 | 0.20 |
| Mycotoxin adsorbent (Mycofix® FUM) | 0.10 | 0.10 |
| Antifungal (MOLD-NIL™ MC DRY) | 0.10 | 0.10 |
| Choline chloride (70%) | 0.10 | 0.10 |
| Antioxidant (OXY-NIL™ RX DRY) | 0.05 | 0.05 |
| **Calculated chemical composition (%)** | | |
| Moisture | 9.36 | 9.36 |
| Crude protein | 36.00 | 36.00 |
| Total lipid | 9.18 | 9.18 |
| Fiber | 3.02 | 3.02 |
| Starch | 25.00 | 25.00 |
| Ash | 8.97 | 8.97 |
| Calcium | 2.36 | 2.36 |
| Phosphorus | 1.40 | 1.40 |
| Digestible energy (kcal kg$^{-1}$) | 3100 | 3100 |
| Digestible protein | 30.93 | 30.93 |
| Total lysine | 2.70 | 2.70 |
| Total methionine + cysteine | 0.95 | 0.95 |
| Tryptophan | 0.39 | 0.39 |
| Threonine | 1.35 | 1.35 |

[1] Premix: vit A 12,000 IU; vit D$_3$ 3000 IU; vit E 150 mg; vit K$_3$ 15 mg; vit B$_1$ 20 mg; vit B$_2$ 20 mg; vit B$_6$ 17.50 mg; vit B$_{12}$ 40 mg; vit C 300 mg; nicotinic acid 100 mg; pantothenic acid 50 mg; biotin 1 mg; folic acid 6 mg; antioxidant 25 mg; Cu 17.50 mg; Fe 100 mg; Mn 50 mg; Zn 120 mg; I 0.80 mg; Se 0.50 mg; Co 0.40 mg; 125 mg inositol; choline 500 mg.

The commercial probiotic AQUA-PHOTO® (Biogenic group, São Paulo, Brazil) composed by *Bacillus subtilis* KCCM10941 ($1.34 \times 10^7$ CFU g$^{-1}$) [11] and *L. plantarum* KCCM11092P ($1.51 \times 10^6$ CFU g$^{-1}$) [21] was mixed in soybean oil (2% of diet weight) and sprayed on feed. The control ration received only 2% of the soybean oil. The rations were stored at 4 °C.

### 2.3.2. Experimental Design

The experiments were carried out using a completely randomized design with five treatments and five replicates per treatment: CON = control (diet feeding, without probiotics and vaccine); ADJ = adjuvant (fish received diet feeding and only adjuvant injection); PRO = probiotics (with probiotics, without vaccination); VAC = vaccine (without probiotics, with vaccination and booster); PRO + VAC = probiotics + vaccine (with probiotics, vaccination, and booster) (Figure 1).

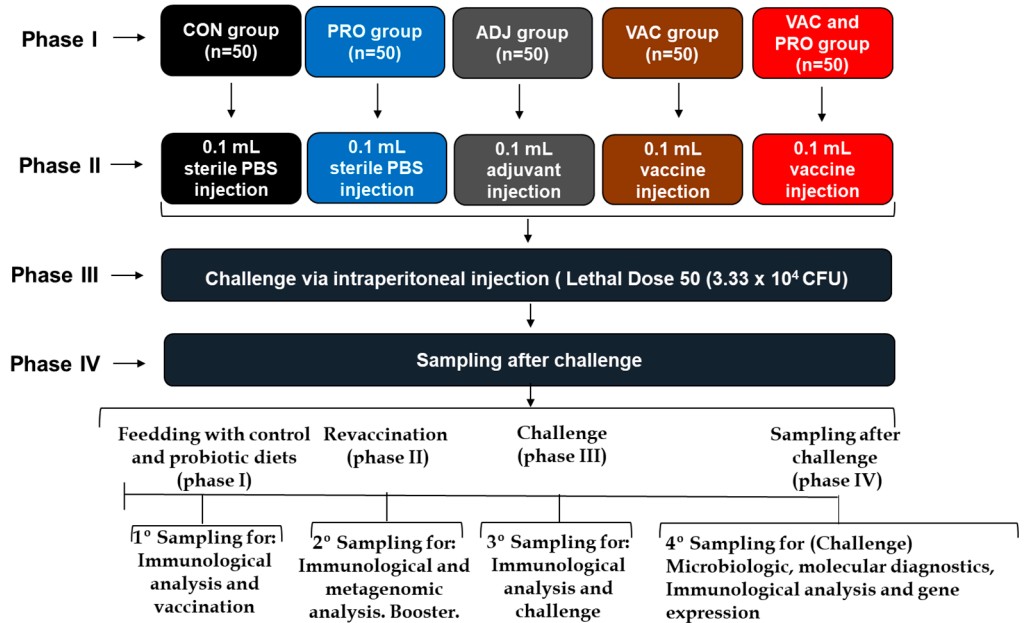

**Figure 1.** Experimental infection scheme of Nile tilapia, *O. niloticus* in vivo assay, fed with probiotic diets, vaccinated (two doses) and challenged against *S. agalactiae* serotype Ib.

A total of 300 Nile tilapia (mean weight 20 g $\pm$ 2.07) were distributed into 25 aquaria of 50 L with water recirculating system, aeration and temperature control. The water parameters (dissolved oxygen level, temperature and pH) were measured by digital oximeter (Hanna® instruments, São Paulo, Brazil), and ammonia and nitrite levels with specific kits (Labcon Test® Alcon company, Santa Catarina, Brazil). The feces were removed three times per week, and photoperiod was adjusted to 12:12 h. The mean and standard deviation of the water quality parameters, for 63 days, were: dissolved oxygen 4.99 $\pm$ 0.10 mg L$^{-1}$; pH 6.79 $\pm$ 0.20; total ammonia 0.27 $\pm$ 0.07 mg L$^{-1}$; nitrite 0.40 $\pm$ 0.20 mg L$^{-1}$; temperature 27.09 $\pm$ 0.10 °C.

This work was divided into phases I, II, III and IV. The fish were fed until apparent satiation with control and probiotic diets (Table 1) during all phases.

In phase I (0–21 days), the fish were fed with diets (Table 1) for 21 days, anesthetized (by immersion with 75 mg L$^{-1}$ eugenol solution), and blood was collected from 10 fish/treatment (CON/PRO). The remaining fish received intraperitoneal injection with: CON and PRO (*n* = 50 fish/treatment) received 0.1 mL of PBS—phosphate buffer saline solution; ADJ (*n* = 50 fish), received 0.1 mL of Marcol-Montanide adjuvant; VAC (*n* = 50 fish), received 0.1 mL of bacterin *S. agalactiae* Ib); PRO + VAC (*n* = 50 fishes), received 0.1 mL of bacterin *S. agalactiae* Ib).

In phase II (21–35 days), fish received the second immunization (revaccination or booster) at day 35 with the same phase I procedures. Before the second immunization, 10 fish per treatment (total *n* = 50 fishes) were randomly selected and anesthetized with eugenol (75 mg L$^{-1}$) to collect blood samples by caudal puncture with non-heparinized syringes for respiratory burst activity assay. Serum samples were collected after blood centrifugation for lysozyme and measurement of IgM. The serum samples of each experimental

group were composed by pools of three or four animals (total of three pools per group). After sampling, the fish were anesthetized again with a high concentration of eugenol (0.10 g L$^{-1}$) and killed by cranial perfusion. The gut portion (anterior portion) samples of 10 fish per treatment were collected, according to [22], and stored in ethanol (99° GL) and −20 °C until proceed the metagenomic analysis at the Microbiology Laboratory, Malaga University (Spain).

In phase III (49–63 days), the infection challenge test proceeded with *S. agalactiae* in the different treatments on day 49, and the fish were observed until day 63. A total of thirty fish from each treatment (six per replicate) were infected with *S. agalactiae* serotype Ib by intraperitoneal injection. During the infection period, the dead fish were collected and maintained at −20 °C for a brain sample to determine the presence of *S. agalactiae* by microbiological isolation in BHI agar plate and molecular diagnostics.

For 14 days, the clinical signs related to streptococcosis (anorexia, lethargy, erratic swimming, exophthalmia and ascites) and the mortality of the animals were observed. The relative protection levels (RPL) were calculated by the following equation: RPL (%) = [1 − (%mortality of treated fish ÷ % mortality of control fish)] × 100 [23]. The dead fish were kept at −20 °C until microbiological and molecular diagnostics.

In phase IV, at the end of the experiment (63 days or four phase), 10 fish per treatment were anesthetized with eugenol (75 mg L$^{-1}$), and blood and head kidney samples were collected for immune-related gene expression analyses (IL-1β, IgM and MHCII). The remaining fish were anesthetized with eugenol (75 mg L$^{-1}$) and killed by cranial perfusion.

### 2.4. Microbiological and Molecular Diagnostics

For re-isolation of *S. agalactiae*, the frozen dead fish with signs of streptococcus during the challenge were maintained at 6–8 °C overnight (approximately 14 h). Afterwards, the brain was extracted with a platinum loop (10 μm) and spread on a BHI agar plate and incubated for 24 h at 30 °C (±1 °C). Isolate multiplication was evidenced by bacterial colony formation. The bacteria morphology was verified by microscopy after Gram staining. The identification of *S. agalactiae* was performed by PCR. For molecular diagnosis, the bacterial DNA was extracted with an E.Z.N.A. bacterial DNA kit (Omega), and PCR was performed with specific primers for *S. agalactiae* (F: CCACGATCTAGAAATAGATTG and R: TGCCAAGGCATCCACC) [17], using 20 μL of the mix composed of the following reagents: 1 × PCR buffer; 2 mM MgCl$_2$; 25 μM dATP, dCTP, dGTP and dTTP; 10 mM primers; and 0.3 U of HotMasterTaq DNA Polymerase. The following steps were carried out to amplify the samples: denaturation (3 min, 95 °C), 36 cycles of 95 °C for 30 s, 58 °C for 40 s and 72 °C for 60 s, and one cycle of 72 °C for 10 min. The negative control utilized ultrapure water, and the positive control was DNA of *S. agalactiae* (PREVET laboratories, Jaboticabal, Brazil). The amplicons were applied to an agarose gel (1.50%) and subjected to electrophoresis.

The identification of bacteria in positive samples was also verified by 16S rRNA gene sequencing [24]. The sequences were edited with the Bioedit program and compared with NCBI data (nucleotide sequences).

### 2.5. Biochemical Analysis

#### 2.5.1. Lysozyme Activity

Lysozyme activity was determined with serum samples, following the method in [25], adapted for a 96-well microplate reader. In each well, we added 200 μL of *Micrococcus lysodeikticus* [1 mg mL$^{-1}$ of lyophilized *M. lysodeikticus* diluted in 0.05 M sodium phosphate buffer (pH 6.30)] and 10 μL of serum sample. The microplate was incubated at 20 °C with shaking, and absorbance was measured (λ = 540 nm) at 0 and 5 min. One unit of lysozyme was considered as the required amount to reduce the absorbance by 0.001 per minute. The results were calculated by the equation: $\Delta ABS \times \frac{1000}{\frac{min}{mL}}$ [26].

2.5.2. Phagocyte Respiratory Burst Activity

Phagocyte respiratory burst activity was determined using nitroblue tetrazolium (NBT), following the method described in [27]. Aliquots of blood (50 μL) and 0.2% NBT (50 μL) were pipetted into tubes and incubated for 30 min at room temperature. Afterwards, in each tube, 1.0 mL of N, N-dimethylformamide (DMF) was added and centrifuged at $3000 \times g$ for five minutes. The supernatant was transferred to cuvettes and absorbance was measured ($\lambda$ = 620 nm).

2.5.3. Assessment of the Specific *S. agalactiae* Antibody Response (IgM)

Thus, 200 μL of *S. agalactiae* bacterin, diluted in carbonate buffer (pH 9.60), at a concentration of $10^7$ CFU, were transferred to 96 well microplate (Nunc MaxiSorp™ Invitrogen by ThermoFisher Scientific, Walthan, MA, USA) and kept at 4 °C overnight. The plates were washed three times with 200 μL with buffer solution (PBS and 0.05% TWEEN 20). Subsequently, 200 μL PBS-Tween containing 3% BSA (Sigma, St. Louis, MO, USA) were added to each well. The plates were incubated for 2 h at 37 °C and then washed. Afterward, 200 μL of each serum sample from the treatment CON, ADJ, PRO, VAC, and PRO-VAC (concentration of 1:20) were pipetted into each well. Two negative controls were used in all plates. The plates were rewashed, and 200 μL of Nile tilapia anti-IgM polyclonal antibody developed in a rabbit (1:1000—in-house produced) were transferred to each well, followed by incubation at 37 °C for 1 h. After the incubation period, 200 μL of anti-rabbit IgG (whole molecule), and peroxidase-conjugated, (1:1000; Sigma, St. Louis, MO, USA) were added and the plates were incubated at 37 °C for 1 h. After three washes, the colorimetric substrate tetramethylbenzidine (TMB) (BD™, Franklin Lakes, NJ, USA) was added to the wells, and the solution was blocked with sulfuric acid at 30%. The absorbance was read at 450 nm [28].

2.5.4. RNA Isolation, cDNA Synthesis, and Quantitative Reverse-Transcription PCR (RTq-PCR)

The head kidney samples (N = 6) of fish at 63 days were removed from the −80 °C freezer and immediately macerated in a mortar containing liquid nitrogen. Total RNA was then isolated from 10 mg of the head kidney using the RNeasy Mini Kit® (Qiagen™, Hilden, Germany), following the manufacturer's protocol. RNA quantity and purity were evaluated with and microplate spectrophotometer (Eon® BioTek, Agilent™, Santa Clara, CA, USA), and quality analyzed with the Agilent 2100 bioanalyzer (Agilent™, Santa Clara, CA, USA). Samples with an RIN value above 7.0 were used to synthesize the cDNA.

The first-strand cDNA synthesis was carried out by reverse transcription (RT) from equal quantities (1.0 μg) of total RNA using the Quanti-nova® (Qiagen™, Hilden, Germany) reverse transcriptase kit (Qiagen™, Hilden, Germany) and oligo (dT) as a primer. The reaction conditions were according to the manufacturer's protocol and included a DNA removal treatment. The first-strand cDNA was quantified using a microplate spectrophotometer, and working solutions were prepared (100 ng μL$^{-1}$) and stored at −80 °C until use in the quantitative reverse-transcription polymerase chain reaction (RTq-PCR).

The determination of gene expression was performed for three genes related to the immune response, MHCII, IgM and IL-lβ, as described in [29–31]. The EF1 and ACTB genes were used as an endogenous control of the reaction [29,30]. Efficiency curves were obtained for each primer, and the values obtained were: ACTB (101%), EF1 (92.9%), IgM (92.4%), MHCII (95.2%) and IL-lβ (91.5 %).

The RT-qPCR reaction mix consisted of 6.25 μL of SYBR® Green (BioRad™, Hercules, CA, USA), 2.5 μL of each primer (10 mM), 2 μL of metagenomic DNA (working solution) and ultrapure water to a final volume of 12.5 μL. The RT-qPCR reactions were performed in a thermocycler CFX96™ Real-Time System (BioRad, Hercules, CA, USA), following the amplification protocol under the conditions: 95 °C for 30 s, followed by 39 cycles of 95 °C for 5 s, 60 °C for 30 s, and a final step of 95 °C for 10 s. At the end of the run, a dissociation protocol was applied to obtain the dissociation curves. The results of the RT-qPCR reactions were analyzed with the BioRad CFX Manager 3.1 program (Applied

Biosystems, Waltham, MA, USA) using the values of the quantification cycle (Cq) according to the $2^{-\Delta\Delta Ct}$ method [32].

### 2.5.5. DNA Extraction, PCR Amplification and 16S rRNA Sequencing

Total DNA was extracted from the gut sample (anterior gut) of 50 fish (10 fish per treatment), following the protocol of [33], modified by [34] (salt precipitation method). For this process, 50 mg of each intestine sample were macerated and homogenized in 300 µL of resuspension buffer (0.10 M Tris-HCl, 0.01 M NaCl, 0.10 M EDTA, pH 8) and 300 µL of lysis buffer (0.1 M Tris-HCl, 0.1 M EDTA, 0.01 M NaCl, 1% SDS, pH 8).

The samples were treated with 20 µL of proteinase K (150 µg mL$^{-1}$) at 55 °C for 2.5 h and lysozyme (10 mg mL$^{-1}$) at 25 °C for 15 min. Next, 192 µL of 6 M NaCl were added to each tube. The solutions were cooled in ice for 10 min and centrifuged at $10,000 \times g$ for 6 min, and supernatants with genomic DNA were transferred to tubes containing an equal volume of isopropanol.

The DNA of each sample was centrifuged at $10,000 \times g$ for 6 min. The supernatant was discarded, and an equal volume of 70% ethanol was added. The tubes were centrifuged at $10,000 \times g$ for 6 min, and the supernatant was removed again. The pellets were resuspended in 100 µL of ultrapure water (DNAse- and RNAse-free) and stored at 4 °C. The concentration and purity were measured using a Qubit 2.0 fluorimeter (Thermo Fisher Scientific, Waltham, MA, USA).

Bacterial 16S rRNA sequencing was performed at the Ultra Sequencing Service of Bioinnovation of Malaga University, using the 2 × 300 bp paired-end sequence of Illumina® MiSeq platform (Illumina, San Diego, CA, USA). Sequencing was initiated with primersa5′aTCGTCGGCAGCGTCAGATGTGTATAAGAGACAGCCTACGGGNGGCWGC AGa3′aanda5′aGTCTCGTGGGCTCGGAGATGTGTATAAGAGACAGGACTACHVGGGTA TCTAATCC 3′, targeting the variables V3-V4 of the 16S rRNA gene. For library construction, the workflow was based on the MOTHUR program (version 1.39.5). First, the adaptor chains were deleted, and final reads were paired, and merged overlay sequences were removed. The reads were filtered according to amplifiers with size variation between 400 and 600 bp. In addition, chimeras and non-specific products from PCR were discarded by the UCHIME program. The sequences were identified in the Greengenes database, and sequences with a similarity of 97% were classified as OTUs (operational taxonomic units).

All data analysis of the intestinal microbiota were processed using the *phyloseq* and *vegan* library in the R statistical package [35]. Rarefaction curves were obtained by plotting the observed OTUs about sequence numbers [36]. Data were normalized based on rarefaction curves, and sequences with less than 10 reads were removed. The Shannon–Wiener, Chao1, and Simpson indices (alpha diversity) were estimated to evaluate the phylogenetic and taxonomic structure diversities.

Principal coordinates analysis (PCoA) of Bray–Curtis dissimilarity was used to generate the ordination of beta diversity of the OTU profiles between the microbiota of each group. The results of the gut microbiological profiles were presented at the following taxonomic level: phylum, class, and genus.

### 2.6. Statistical Analysis

Data of lysozyme and respiratory burst activities were submitted to a homoscedasticity test and analysis of variance (ANOVA), and means were compared by Tukey's test ($p < 0.05$), using SAS 9.1 software. The results for survival rate were analyzed in Microsoft Excel 2016, while GraphPad Prism version 7 (GraphPad Software Inc., San Diego, CA, USA) was used to conduct the Kaplan–Meier analysis [37] paired with subsequent Mantel–Cox log-rank tests applied to mortality data to calculate survival probabilities and compare survival distributions of fish in treatment versus the control, and measurement of IgM with Real-statistic-Excel with descriptive statistics and normality, ANOVA/single factor/ Kruskal–Wallis/ Dunn, comparing the treated and control. The gene expression data were

evaluated by the $2^{-\Delta\Delta Ct}$ method, considered as an adjusted form to determine the relative changes related to gene expression [32].

All data analysis of the intestinal microbiota was processed using the normality test (Shapiro–Wilk). Variance homoscedasticity (Levene) of the alpha diversity data were determined; ANOVA was applied to detect differences between the alpha diversity data and means compared with the Tukey test ($p < 0.05$).

To see if these differences in PCoA were significant, the dissimilarity matrices obtained with the Bray–Curtis index were evaluated by permutation multivariate variance analysis (PERMANOVA) with 999 permutations using the PAST software [38] version 3.16. The relative abundance data were submitted to ANOVA and the means compared by the Tukey test ($p < 0.05$).

## 3. Results

### 3.1. Vaccine Efficacy

The Nile tilapia vaccine immunization with probiotic administration provided significantly more protection against *S. agalactiae* infection (survival rates of 87% for VAC and 97% for PRO + VAC vs 40% for CON) (Figure 2, Table 2). The relative protection level of VAC (78%) and PRO + VAC (94%) was higher compared to the ADJ (28%) and PRO (44%) treatment. (Table 2).

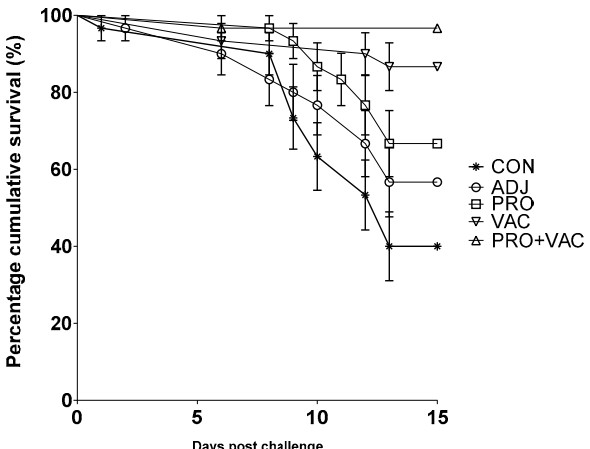

**Figure 2.** Accumulated Nile tilapia (*O. niloticus*) survival rate during 14 days of *S. agalactiae* challenge infection (CON: control; ADJ: adjuvant; PRO: probiotics; VAC: vaccinated; PRO + VAC: probiotics + vaccine).

### 3.2. S. agalactiae Isolation after Infection Challenge Test

Gram staining and microscopic evaluation confirmed Gram-positive cocci of isolated from fish brain samples. The PCR analyses of twenty-one samples confirmed the *S. agalactiae*. Analysis by agarose gel electrophoresis showed molecular weight of approximately 272 bp. The 16S rRNA sequencing analyses confirmed the identity of the isolates as *S. agalactiae* [24].

### 3.3. Phagocyte Respiratory Burst and Lysozyme Activities

Lysozyme and phagocyte respiratory burst activities did not differ between treatments at any time evaluated (Supplemental Figure S1).

**Table 2.** Accumulated fish mortality during *S. agalactiae* infection challenge of 5 aquarium with 6 fish/tank/challenge group and relative protection level (RPL%) of homologous vaccine. Significant differences with control, by the Mantel–Cox log-rank test: (probability, $p < 0.05$).

| Treatments | Days Post Infection | | | | | | | | RPL |
|---|---|---|---|---|---|---|---|---|---|
| | **2** | **3** | **4** | **5** | **6** | **7** | **9** | **14** | |
| Control (CON) | 4 | 7 | 7 | 10 | 15 | 17 | 17 | 17 | - |
| Adjuvant (ADJ) | 4 | 7 | 7 | 8 | 8 | 10 | 10 | 13 | 28% |
| Probiotics (PRO) | 2 | 4 | 5 | 7 | 8 | 9 | 9 | 9 | 44% |
| Vaccine (VAC) | 1 | 2 | 2 | 2 | 2 | 2 | 4 | 4 | 78% |
| Probiotics + Vaccine (PRO + VAC) | 0 | 0 | 0 | 0 | 0 | 0 | 1 | 1 | 94% |

$n$ = 30/treatment (number of fish per challenge group).

### 3.4. Quantification of IgM Produced after Probiotic Diet Feeding, Vaccination and Infection Challenge Test

In phase I (21 days of feeding) fish from CON have more IgM than PRO ($p = 0.002$). At 35d (phase II), VAC showed higher IgM production when compared to other (CON, ADJ, PRO and PRO + VAC) by Kruskall–Wallis test ($p = 0.001$) and Dunn test ($p > 0.05$) (Figure 3).

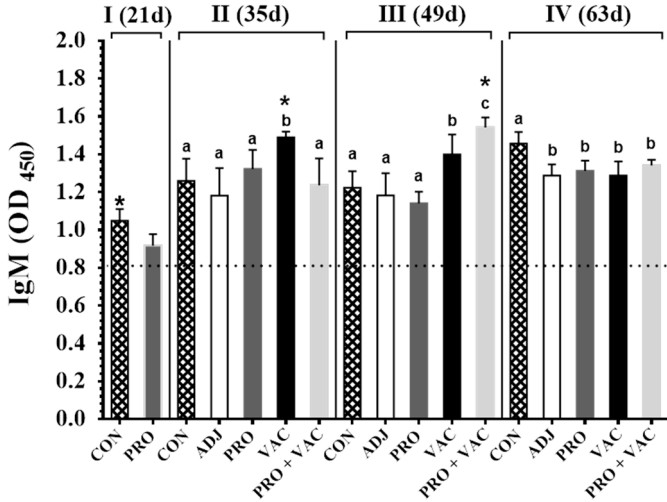

**Figure 3.** ELISA indirect for measurement IgM in serum of Nile tilapia collected in the periods: [I] probiotic diet feeding, [II] vaccination, [III] revaccination (booster) and [IV] survival post-infection with *S. agalactiae* serotype Ib. Statistical differences between control and treated groups ($p < 0.05$). (CON: control; ADJ: adjuvant; PRO: probiotics; VAC: vaccinated; PRO + VAC: probiotics + vaccine). * Significant difference in period II Kruskall–Wallis ($p > 0.001$) and period III, IV Kruskall–Wallis ($p > 0.001$). [a–c] Significant difference in Dunn test between groups within each period. $OD_{450}$: Optical Density at 450 nm.

After 49 days (phase III), the fish treated with PRO + VAC showed more IgM than compared to VAC and controls. The Kruskall–Wallis test showed that difference between groups ($p > 0.0001$) and Dunn showed differences between CON/VAC; CON/PRO + VAC; ADJ/VAC; ADJ/PRO + VAC; PRO/VAC; PRO/PRO + VAC.

Comparing fish from VAC (35d phase II) with VAC (49d phase III) there was less IgM (0.091 OD) than PRO + VAC (35d) PRO + VAC (0.305 OD).

At the end of experiment (63d) fish showed IgM statistical differences between the CON with all treatments.

*3.5. Gene Expression*

At 49d with *S. agalactiae*, the expression of IgM, MHCII and IL-1β genes of head kidney samples did not differ between treatments (Figure 4).

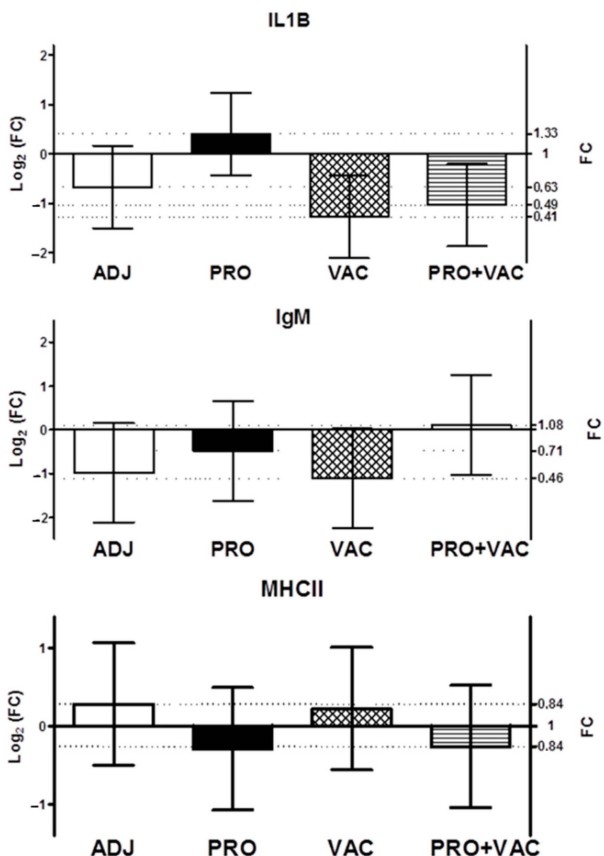

**Figure 4.** Relative expression of pro-inflammatory and immune-related gene in head kidney samples of Nile tilapia (*O. niloticus*), survivors from 14 days post-infection (dpi) with *S. agalactiae* serotype Ib. (CON: control; ADJ: adjuvant; PRO: probiotics; VAC: vaccinated; PRO + VAC: probiotics + vaccine).

3.5.1. Metagenomic Analyses

A total of 5,095,041 validated sequences were obtained for both forward and reverse directions after sequencing with Illumina Miseq. The mean read depth per sample was 113,223.13 ± 24,156.19 (mean ± SD) sequences per read, and 57,062 readings were filtered according to the rarefaction curve. The rarefaction curve (Figure S2), tended to approach the saturation plateau with good coverage coefficient >99%, indicating adequate sequencing depth. The sequences with less than 10 counts were removed from the taxonomy results, which showed as 549 OTUs obtained through sequencing, with similarity above 97% at the genus level based on the Greengenes database.

3.5.2. Alpha and Beta Diversity

The alpha diversity indices calculated from gut microbial profiles of Nile tilapia (Chao1, Shannon and Simpson) did not show significant differences between treatments $p > 0.05$ (Table 3).

**Table 3.** Alpha diversity indices (Chao 1, Shannon, and Simpson) of gut bacterial communities of Nile tilapia from following groups: CON = control; ADJ = adjuvant; PRO = probiotics; VAC = vaccinated; PRO + VAC = probiotics + vaccine.

| | Alpha Diversity Index | | |
|---|---|---|---|
| **Treatment** | **Chao1** | **Shannon** | **Simpson** |
| CON | 599.54 ± 23.14 | 4.79 ± 0.23 | 0.97 ± 0.01 |
| ADJ | 605.48 ± 52.24 | 4.79 ± 0.31 | 0.97 ± 0.01 |
| PRO | 598.97 ± 90.22 | 4.69 ± 0.25 | 0.97 ± 0.01 |
| VAC | 566.71 ± 130.22 | 4.59 ± 0.65 | 0.96 ± 0.05 |
| PRO + VAC | 572.08 ± 82.47 | 4.86 ± 0.38 | ± 0.01 |

Beta diversity analysis was illustrated as a PCoA plot according to the microbiota composition of the gastrointestinal (GI) tract of specimens studied at the OTU level (Figure 5). PERMANOVA according to the OTUs and based on Bray–Curtis index did not show significant differences in microbiota composition between the samples analyzed ($p > 0.05$) (Table 4).

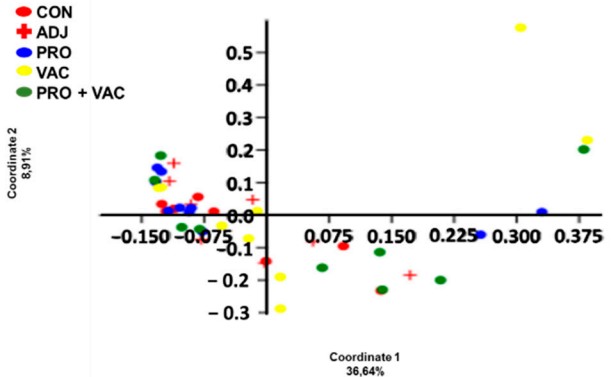

**Figure 5.** Principal coordinates analysis (PCoA) plots of the GI tract microbiota composition studied at the OTU level, from the Nile tilapia gut bacterial community (CON: control; ADJ: adjuvant; PRO: probiotics; VAC: vaccinated; PRO + VAC: probiotics + vaccine).

**Table 4.** Permutational multivariate analysis (PERMANOVA) of OTU derived variance from gut bacterial communities of Nile tilapia. Tests were based on Bray–Curtis index dissimilarity distances and 999 permutations.

| | PERMANOVA | | | | |
|---|---|---|---|---|---|
| | **CON** | **ADJ** | **PRO** | **VAC** | **PRO + VAC** |
| **CON** | - | 0.746 | 0.371 | 0.353 | 0.173 |
| **ADJ** | 0.746 | - | 0.906 | 0.737 | 0.48 |
| **PRO** | 0.371 | 0.906 | - | 0.477 | 0.28 |
| **VAC** | 0.353 | 0.737 | 0.477 | - | 0.847 |
| **PRO + VAC** | 0.173 | 0.48 | 0.28 | 0.847 | - |

### 3.5.3. Gut Microbiological Profiles

The OTUs of six different phyla with abundance values over 1% of total gut microbiota were represented (Figure 6A). The predominant phyla (>1%) detected in all samples of this study were Actinobacteria, Bacteroidetes, Firmicutes, Fusobacteria and Proteobacteria. Fusobacteria was the most abundant phylum in all treatments, but vaccinated fish (VAC) and fish fed with probiotic diet and vaccinated (PRO + VAC) showed significantly lower percentages 52.57 and 63.93%, respectively ($p < 0.05$), whereas fish fed with probiotic (PRO) showed highest the abundance in this phylum (76.42%) between treatments.

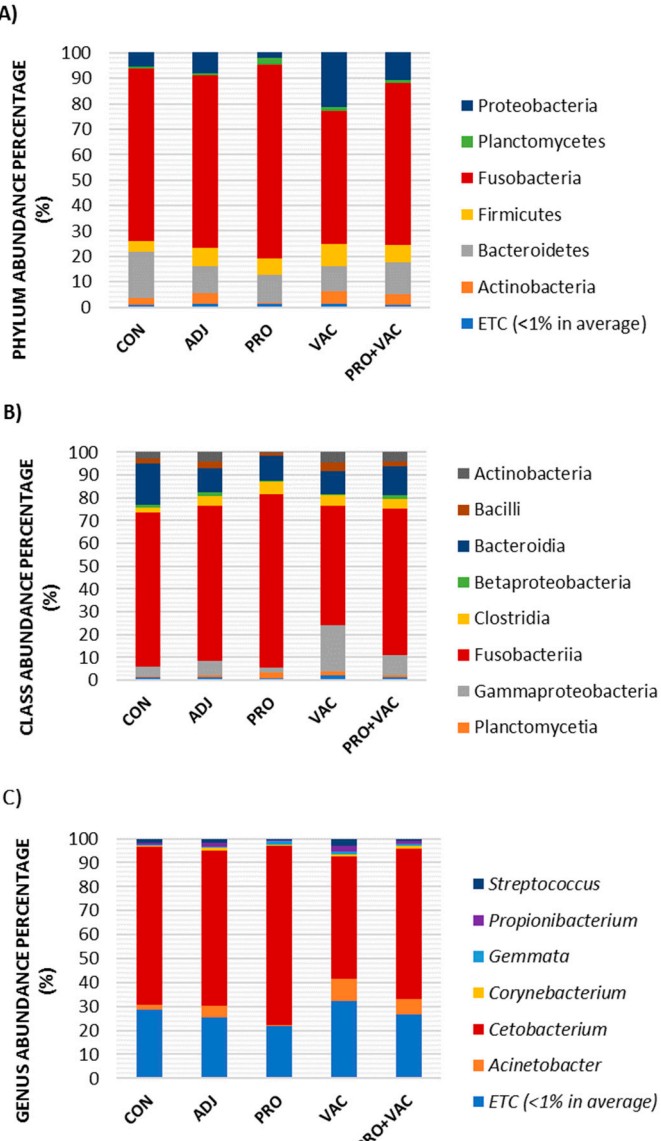

**Figure 6.** Intestinal microbiota composition of (OTUs, operational taxonomic units) of phylum (**A**), class (**B**), and genus (**C**) from Nile tilapia groups: CON = control; ADJ = adjuvant; PRO = probiotics; VAC = vaccinated; PRO + VAC = probiotics + vaccine).

Vaccinated fish also showed the highest occurrence of Proteobacteria (21.97%) compared to other treatments: CON (6.12%), ADJ (9%), PRO (3%) and PRO + VAC (11.54%) ($p < 0.05$).

The abundance of the phylum Bacteroidetes was significantly higher in the control (18.12%) (Figure 7A). At the class level, the community structure and abundance observed were Actinobacteria, Bacilli, Bacteroidia, Betaproteobacteria, Clostridia, Fusobacteria, Gammaproteobacteria, and Planctomycetia among the treatment (Figure 6B). The most predominant class in all treatments was Fusobacteria; animals from the PRO showed the significantly highest Fusobacteriia abundance index, whereas VAC and VAC + PRO the lowest significant abundance with respect to this class (Figure 7B).

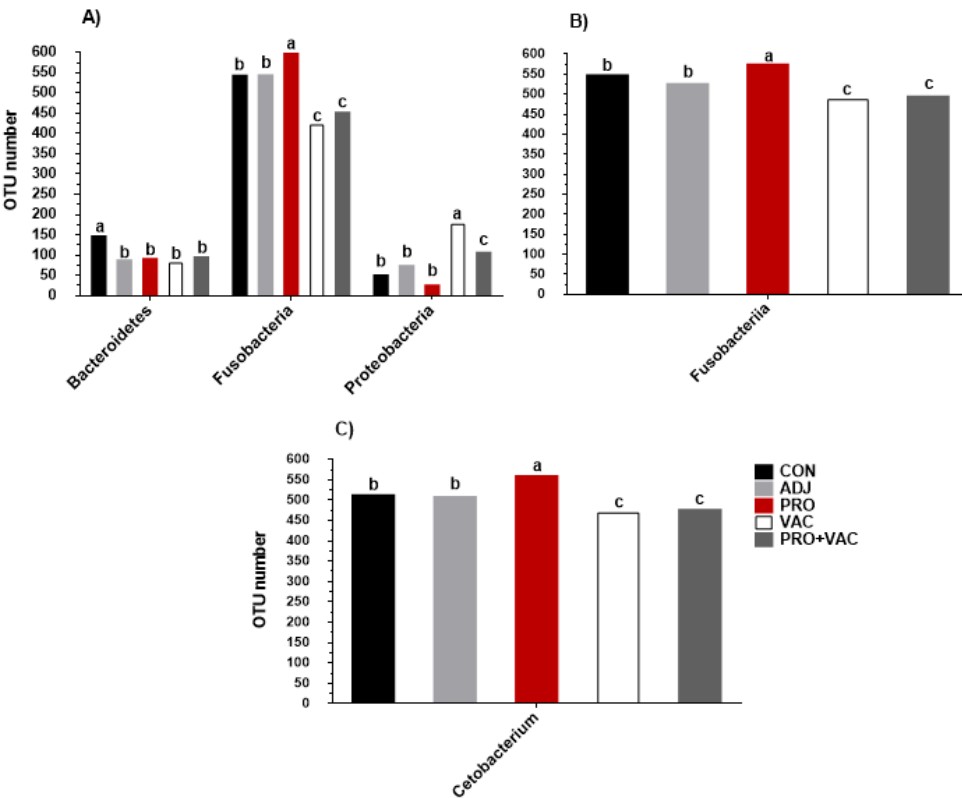

**Figure 7.** Taxonomic differences between treatments. Comparison of relative abundance at the bacterial phylum (**A**), class (**B**) and genus (**C**) level detected in Nile tilapia gut from the following groups: CON = control; ADJ = adjuvant; PRO = probiotics; VAC = vaccinated; PRO + VAC = probiotics + vaccine. [a–c] Different letters indicate significant differences ($p < 0.05$) between mean values of treatments by ANOVA and post hoc Tukey–Kramer test.

The genera observed in Nile tilapia intestine samples with abundance >1% were *Streptococcus*, *Propionibacterium*, *Gemmata*, *Corynebacterium*, *Cetobacterium*, and *Acinetobacter*. *Streptococcus*, *Propionibacterium*, *Acinetobacter*, and *Cetobacterium* were found regardless of treatment. The last taxon showed the highest abundance percentages in the treated and the control (Figure 6C). Moreover, animals fed with probiotic diets showed the highest index of *Cetobacterium* abundance (74.78%), and the vaccinated fish, VAC and PRO + VAC, significantly demonstrated the lower abundance values, 51 and 62.76%, respectively (Figure 7C). Cetobacterium, belonging to the Fusobacteria class, was the main genus that contributed to the significant differences in the gut microbiota composition of animals fed with probiotics when compared to other treatments (Figure 7C).

## 4. Discussion

### 4.1. Experimental Infection with S. agalactiae

Intensive production of Nile tilapia increases the disease outbreaks occurrences and there is limited information on probiotics with vaccination utilization [39,40]. In this study, the fish which received a probiotic diet and vaccination had the highest survival rates (97% at 14 dpi). This synergistic effect between the vaccine and probiotics on fish survival rate can improve the animal resistance to pathogens due to humoral and non-specific immune responses.

Our results demonstrated high levels of IgM after the first immunization, mainly in the VAC, according to [41] animals vaccinated with *S. agalactiae* or challenged demonstrated high levels of IgM. The second immunization (revaccination) was effective in maintaining high levels of IgM. Interestingly, the PRO + VAC in 49 days had a higher level of IgM than VAC at the same time. In analyzing the ability of antibodies to protect against *S. agalactiae*

infection (Figure 3, 63 days), we observed the difference in antibody levels between ADJ, VAC and PRO + VAC groups. The successful connection between the innate and the adaptive immune systems may improve fish resistance against bacterial infection [42].

The synergistic effect of vaccination and oral probiotic administration has also been shown by other infection studies which observed a better survival rate and relative protective level in Nile tilapia fed probiotic diets (Organic Green™ and Vet-Yeast™) and vaccinated against *A. hydrophila* [43]. The authors relate this positive result with the probiotics' effect to stimulate phagocytic activity (Organic Green™), complement activity, and interleukin 1 (IL-1β) gene expression, promoted by β-glucans (Vet-Yeast ™). In another study, the immunomodulatory capacity of probiotic *Lactobacillus sporogenes* in Nile tilapia, orally vaccinated against *Aeromonas hydrophila*, improved the adaptive immune responses when compared to the vaccinated and not supplemented with probiotics [44].

The fish from PRO had a higher survival rate (67%) than the control (40%) after *S. agalactiae* infection and this can be attributed to the probiotic *L. plantarum* added to Nile tilapia diet and it may stimulate the innate immune response. The inclusion of *L. plantarum* in the diet of rainbow trout (*Oncorhynchus mykiss*) improved the survival rate after infection with *Lactococcus garvieae*. According to [45], the higher expression of pro-inflammatory cytokines (IL-1β and TNF-α) in the head kidney of probiotic-fed fish before infection may be associated with upregulation of the anti-inflammatory cytokine IL-10 at 63d. The same probiotic promoted upregulation of IL-8 (CXCL-8, neutrophil chemotactic factor) and IgT genes in the intestine is probably due to positive correlation between the lactic acid bacteria and GALT modulation [16].

Inactivated vaccine induces the adaptive immune response and produces high-affinity specific antibodies, promoting suitable protection against infectious diseases [46]. There is a high correlation between specific antibodies (anti-*S. agalactiae*) concentration and low mortality rate by streptococcosis, corroborating with fish IgM triter elevation and survival rate (87%) in this work [47].

The VAC showed higher levels of IgM in 35d than PRO + VAC treatment. In 49 day fish from PRO + VAC presented more IgM production than VAC. The *Labeo rohita* fed with ration containing $10^{10}$ CFU g$^{-1}$ of *B. subtilis* KADR1 and immunized with cellular components (*B. subtilis* KADR1) showed higher serum IgM levels than control diet [48].

The RPL improvement of the PRO + VAC demonstrated the beneficial interaction between probiotic feeding and vaccination. The probiotic *L. plantarum* may stimulate the maturation of dendritic cells, promoting the polarization of CD4$^+$ T cells towards Th2 response, thereby enhancing protection against *S. agalactiae* infection by inducing the production of cytokines and stimulating B lymphocytes in the humoral immune response. The combined stimulation of both innate and adaptive immune responses could have occurred in PRO + VAC individuals [49].

At 63d, we did not detect any difference in gene expression of MHCII, IL-1b and IgM between treatments. Other authors [46] have shown evidence of upregulated expression of immunological genes up to 72 h after infection. Thus, challenge trials in future studies may consider the initial period after pathogen inoculation.

This study has a limitation regarding to the unchallenged control group absence of experimental bacteria challenge design. The LD50 previously conducted was used to validate the non-mortality of unchallenged group.

### 4.2. Relationships among the Gut Microbiological Profiles of Different Groups

Extrinsic factors (nutrition, health, and the environment) influence the intestinal microbial composition of the animal [50]. The intestinal microbiota has an essential function that supports the epithelial barrier, inhibits the superficial adhesion of pathogens, suitable modulation, maturation of gut immune system and short-chain fatty acids' metabolite release [51]. For aquatic organisms, the constant exposure to various microorganisms, including pathogens and opportunists, may colonize the internal and external surfaces of the animal [7,50,52].

The non-significant differences of biodiversity indices and PCoA data obtained in this study between treatments indicate that the effect of giving probiotics supplementation for 49 days, and vaccination, can be associated only with the relative abundance of bacterial species in the gut microbiota [53].

We detected a similar number of phyla (Actinobacteria, Planctomycetes, Bacteroidetes, Firmicutes, Fusobacteria, and Proteobacteria) among the treatments, which are generally found in freshwater fish gut microbiota [50,52,54]. This may be related to probiotics effects that may variate between species, stages, time of administration and dose [9].

However, we emphasize that the oral administration of probiotics can affect the intestinal microbiota, causing positive effects on commensal bacteria proliferation and reducing the pathogenic species abundance [55–57]. In this work, intestinal microbiota modulations were observed, mainly in fish fed with *L. plantarum* and *B. subtilis*, which resulted in higher percentages of Fusobacteria, probably correlated with *Cetobacterium* values ($p < 0.05$) (Figure 7).

The genera *Cetobacterium* synthesize vitamin B12 and inhibit pathogenic bacteria, improving the immune response, nutrient digestibility, and growth performance of the host [52,58]. According to [59], the oral administration of *Shewanella xiamenensis* A-1, *Aeromonas veronii* A-7, and *B. subtilis* to grass carp (*Ctenopharyngodon idella*) increased the abundance of *Cetobacterium* and enhanced the immune system.

In this study, fish vaccinated intraperitoneally (VAC and PRO + VAC) showed some different results regarding the abundance index, when compared to the CON and PRO, mainly, a lower percentage of *Cetobacterium* and highest index of Proteobacteria.

Therefore, we suspected the increased Proteobacteria numbers in vaccinated groups might have been related to the presence of bacterin [53], the adjuvant group did not show similar results. The authors of [60] observed the symbiotic interaction between microbiota and intestinal epithelial cells, glycoproteins (mucins), antibacterial molecules such as $\alpha$-defensins, type C lectins, lysozyme, phospholipase A and immunoglobulins (secretion). Moreover, [61] inferred that probiotic supplementation preserves intestinal structure and functions due to its inhibitory action against pathogens and stimulation of innate responses.

The microbiome composition impact on vaccination immunogenic capacity is supported by [50], and other authors have demonstrated the increase in total IgM titer ($4.3 \pm 0.22$ mg mL$^{-1}$) of rainbow trout (*O. mykiss*) fed with probiotic composed of *Lactobacillus plantarum* 426,951 and vaccinated against *Yersinia ruckeri* [62].

Therefore, this study showed that a combination of probiotic feeding with vaccination is an effective and viable strategy to control streptococcosis outbreaks in fish production systems, thereby avoiding economic losses due to mortality.

## 5. Conclusions

The commercial probiotic AQUA PHOTO® composed of *B. subtilis* and *L. plantarum*, administered orally in Nile tilapia, and the vaccination against *S. agalactiae* increased the survival after homologous strain infection challenge and modified the composition of the host intestinal microbiota.

**Supplementary Materials:** The following supporting information can be downloaded at: https://www.mdpi.com/article/10.3390/fishes7040211/s1, Figure S1: (A) Lysozyme activities and (B) Phagocyte respiratory burst of Nile tilapia, *O. niloticus*, at phases: [I] probiotic diet feeding [II], vaccination, [III] revaccination (booster) and [IV] survivors from 14 days post-infection (dpi) with *S. agalactiae* serotype Ib. (CON: control; ADJ: adjuvant; PRO: probiotics; VAC: vaccinated; PRO + VAC: probiotics + vaccine); Figure S2: Rarefaction curves of OTUs (operational taxonomic units), grouped in sequences with over 97% similarity, from gut bacterial community of Nile tilapia (CON: control; ADJ: adjuvant; PRO: probiotics; VAC: vaccinated; PRO + VAC: probiotics + vaccine).

**Author Contributions:** M.C.G.: Conceptualization, Methodology, Investigation, Formal analysis, Writing—original draft, Writing—review, editing and Funding acquisition. M.F.F.-A. and M.M.N.: Investigation, Formal analysis, Visualization, Validation, Supervision, Writing—review & editing.

L.Y.S. and C.A.T.K.: Formal analysis, Visualization, Validation. I.M.C., M.A.M. and S.T.-P.: Investigation, Writing—original draft, Writing—review & editing. D.d.C.D. and C.M.I.: Conceptualization, Methodology, Investigation, Formal analysis, Visualization, Validation, Writing—original draft, Writing—review & editing. L.L.C.: Formal analysis, Visualization, Validation. E.E.B., P.B.C. and D.P.O.: Methodology, Investigation, Formal analysis, Visualization, Validation. M.J.T.R.-P. and L.T.: Conceptualization, Supervision, Project administration, Writing—review & editing, Funding acquisition. All authors have read and agreed to the published version of the manuscript.

**Funding:** This work was supported by FAPESP (Process No. 2019/20441-8, 2017/23225-9 and 2017/03738-1) and Coordenação de Aperfeiçoamento de Pessoal de Nível Superior—Brasil (CAPES).

**Institutional Review Board Statement:** The experiments were performed at the Aquaculture Research Center/Fisheries Institute/APTA/SAA, São Paulo, Brazil). The experimental infection assay was carried out according to Resolution No. 592 of the Brazilian Federal Council of Veterinary Medicine and precepts of Ethical Principles of Animal Experimentation. This project was approved by the Fisheries Institute Ethics Committee (protocol No. 01/2018).

**Informed Consent Statement:** Not applicable.

**Data Availability Statement:** Not applicable.

**Acknowledgments:** We thank company Biogenic for commercial probiotic donations and Biocamp for vaccine production and supply. Furthermore, our thanks go to Carolina Perico Graciano, Rodrigo Hozana Ferreira, Sabrina França Lopes, Diego de Souza Vicente, Edmilson Zanfurlin Lima and Claudio Cirineu Ciola for technical support. A. Leyva (USA) provided English editing of the manuscript. We are also grateful to Jorge García Márquez and Marta Domínguez Maqueda for technical support and ultra-sequencing Service of Bioinnovation Center, Malaga University for metagenomics analysis support.

**Conflicts of Interest:** All authors contributed significantly to the manuscript and are in agreement with its content. No authors have any potential financial conflicts of interest related to this manuscript.

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
