# Peer review of "Oral Administration of Probiotics (Bacillus subtilis and Lactobacillus plantarum) in Nile Tilapia (Oreochromis niloticus) Vaccinated and Challenged with Streptococcus agalactiae"

_fishes, doi:10.3390/fishes7040211_

Round 1

Reviewer 1 Report

Authors have improved the manuscript significantly in regard to methodological description and discussion quality.

However, I still have some concerns, which have not been addressed from the previous submission, but response to the first review was unfortunately not provided by the editorial office.

Language quality has been improved, and is adequate specifically in the new discussion section, but the other sections still need some language editing. Some examples follow.

L46: either “results, which were” or better “production levels, which reached over…”.

L54: either “bacteria entering the intestinal tract, cross physical and chemical barriers” or “bacteria enter the intestinal tract by crossing”.

L55: “The infection can result in systemic dissemination”

There are pending issues from the previous manuscript version on the IgM ELISA. First, authors are NOT measuring total IgM levels from their serum samples, but SPECIFIC IgM. When wells are first coated with the specific bacterial antigen (S. agalactiae bacterin), after blocking, only specific anti-S. agalactiae IgM in fish serum binds to the only available S. agalactiae epitopes. The remaining IgM molecules in the sera, which are not anti-S. agalactiae, do not bind and are washed away. Second, the used ELISA technique does not quantify IgM. The results are given as O.D. (not mg/ml), and they allow comparison between sample titers, i.e. if a sample contains more or less anti-S. agalactiae IgM than another one. Authors cannot talk about IgM quantification or total IgM levels of their samples and should correct L168, L231, L328, L383, L391, L537.

Statistics in Figure3 are still confusing. Apparently, asterisks and letters are representing the same comparisons (between experimental groups within a period) performed by different tests, which is unnecessary. Why have authors not compared IgM levels of each individual group at the different times?

L554-555: The stimulation of an innate immune response was not found by the authors (lysozyme, respiratory burst, IL1b) in any experimental group. Although circulating specific IgM responded significantly in VAC and PRO+VAC pointing to an adaptive immune response elicited by vaccination, IgM expression and MHCII transcripts in head kidney did not support this result. Authors should address these issues.

Further minor issues:

- L72: Write the entire name of the bacterial species on their first mention.

- Be consistent in the use of the right symbol for degrees (°, not º).

- L143: Authors used one vaccine dose (3.33 x 10E8 CFU/mL), but applied it twice.

- L173: define GL

- L240: PRO treatment is missing

- Be consistent with P values: P, p

- Table2 contains core results of the experiment and this data greatly improve manuscript quality. Legend is however confusing, I would say “Accumulated mortality in the different groups of fish after challenge and average survival (% +/- Standard deviation (SD)) of …”. Additionally, dpi and RPL descriptions are missing and “P=probability” is superfluous.

- L363-364: why 21 samples? At 13 dpi dead fish should be 17+13+9+4, so 43 samples.

- L395 & 396: P>0.001 or P<0.001

- L438: define GI

- Figure8B): If mean values are represented, why is SD not shown? Fusobacteria under the x axis needs correction

Reviewer 2 Report

In this new version of the manuscript entitled “Oral administration of probiotics (Bacillus subtilis and Lactobacillus plantarum) in Nile tilapia (Oreochromis niloticus) vaccinated and challenged with Streptococcus agalactiae”, the authors have considerably improved the quality of the presentation of their work, considering some of the comments of the reviewers.

Even if the authors presented several interesting results, some aspects of their manuscript could be improved:

-       Why did the authors not include a control group non-exposed to the pathogenic bacterium? This information will provide the reader with a good background about the basal immunological level of the fish as well as the resilience of the microbiota of each challenged group at the end of the infection experiment. With this experimental design, the authors cannot rule out the changes in gut microbiota induced by GBS infection. Please, clarify this point.

-       Since probiotics could have an impact on fish growth and development, did the authors evaluate the impact of this treatment on fish growth performance?

-       For immune response analysis, why do the authors focus on these three genes (MHCII, IgM and IL-1ß)? are they related specifically to S. agalactiae-induced infection immune response?

-       Figure 3: statistics are confusing, and it is not clear why time dynamics have not been statistically analyzed (differences between means of the same diet group along the experimental timepoints).

-       For microbiota composition analysis, mainly for taxonomical differences analysis, I strongly recommend using well-adapted statistical methods such as LefSE or ANCOM, instead of comparing only the most abundant taxonomic groups. In the way in which the authors have analyzed their microbiota composition data, they cannot show if there are any OTU specific to an experimental group.

-       Figure 7: instead of representing the average relative abundance of each taxonomic group, I strongly recommend representing the taxonomic composition of each sample in order to better show if there is or not inter-individual variability or not. This information will clearly show how representative are these averages values compared to each sample.

-       In the Discussion section, the authors hypothesize that the increased survival observed in the PRO group could be due to the immune response induction induced by the probiotics. However, what is known about anti-GBS activity by any of the bacterial species forming the AQUA PHOTO commercial probiotic? The antibacterial activity should not be ruled out.

-       In the literature, there are some works describing Nile tilapia microbiota composition. How similar or different is the microbiota of the current work compared to these other works? Please, further analyze and describe this point.

-       The Discussion section is quite long and confusing, with some redundant information. I recommend shorting it to make it easier for the reader the comprehension of the main(s) message(s) of the manuscript, also avoiding unnecessary speculations (i.e. lines 549-553). Idem for the Abstract section.

-       At the end of the Discussion section (lines 629-633), the authors stated that prophylactic protocols combining antibiotics and vaccination should consider the adverse effects of medicaments on gut microbiota profile, and that changes in normal microbiota composition may have an important impact in fish health homeostasis. However, since with their results they are also showing changes in microbiota composition induced by the combination of probiotics administration and vaccination, why they strategy is more adequate that the other?

-       Please, verify the bibliography because there are some citations that there are not correctly referenced in the text (i.e. Hao et al., 2017).

 Some minor comments:

-       Lines 90-92: this information is quite redundant with the paragraph on lines 59-66. Please, try to avoid repetitions.

-       Lines 100-101: instead of “In this work… preparation” I suggest just saying “the DL50 for this bacteria strain as previously determined (23). Also remove the sentence on lines 102-103.

-       Line 106: is this bacteria concentration correct? If yes, why this concentration a not other?

-       I recommend merging sections 2.3.2 and 2.3.3.

-       Figure 1 quality should be improved and simplified since it is not easy to follow.

-       Line 145: please, include the standard deviation for the fish mean weight.

-       Line 542: “Labeo” instead of “Lobeo”.

-       Line 590: “Figure 8” instead of “Figure 2”.

Round 2

Reviewer 2 Report

I thank the authors for their exhaustive work in answering and including all the modifications suggested during the review of their manuscript. Even if I can fully understand the limitations concerning the number of aquaria, conditions, etc. the authors should include a couple of sentences explaining the limitations of their study, such as the lack of non-infected control groups.

During the revision process, the authors mentioned several times in the text “anti-S. agalactiae IgM”. Even if I understand that this modification was to answer one of my previous questions, this is not correct since IgM levels could be affected by many different pathogenic microorganisms. Instead of this formula, I strongly recommend justifying once why they measure IgM-levels (including bibliographic references demonstrating that S. agalactiae increases its serum levels, and then just speaking about IgM.

Concerning my question 6, about representing the taxonomic composition of each sample to better show if there is or no inter-individual variability or not, the authors answered that this information is included in Table 8. However, there is no table 8 in the text. Please, verify.

Also, since it has been previously described the anti-S. agalactiae activity of AQUAPHOTO probiotics, the authors should include this information and the corresponding reference in the text.

There is also a problem with the answer to question 8, because the authors answered that the requested information is on lines 583-625. However, these lines correspond to the conclusions and Authors' contribution. Please, verify.

There is still a problem with the bibliography format, since there are some references ( in lines 562-567) in which the referenced papers are mentioned without any associated number. Please, verify again the bibliography format.

Ref 51 is not the best one to support the functions of probiotics. I suggest changing it for another reference more adapted to the aquaculture domain.

Author Response

This manuscript is a resubmission of an earlier submission. The following is a list of the peer review reports and author responses from that submission.

Round 1

Reviewer 1 Report

Authors have studied the immune response, intestinal microbiota composition and survival to Streptococcus agalactiae challenge in Nile tilapia upon dietary probiotic administration of Bacillus subtilis and Lactobacillus plantarum and vaccination against the bacterial offender.

Although the experimental fish group treated with probiotic and vaccine showed a promising response upon bacterial challenge, the manuscript presents important weaknesses and cannot be published.

  • Language quality along the entire manuscript requires extensive improvement. A native speaker should rewrite the text.

  • The materials and methods section is especially confusing, since many methodological questions arise.

Eg.       - Time sequence in Figure 1 is confusing and different from the explained timing in L157-160.

            - Fasting time is not mentioned and influences the gut microbiota.

            - Numbers of sampled fish are not clearly indicated and neither are numbers of analysed samples. Authors indicate once the pooling of blood samples, but it remains unclear if pooling is done in every sampling and how many samples were analysed for each technique and each experimental timing.

            - No growth performance was analysed, although fish were subjected to dietary treatments. At least biometric data should have been registered.

            - Head kidney samples were taken after the challenge but a control unchallenged group is missing for comparison.

  • Results are not clearly presented.

            - Results from the macroscopic examination of pathological alterations (L198-199) are missing.

            - Results from statistic analysis on survival/mortality data are missing.

            - Numbers of surviving fish for each dietary treatment are not indicated and it is unclear where the 21 fish mentioned in L365 come from.

            - Non significant results of respiratory burst and lysozyme activities are missing.

            - In Figure 3, shown statistics are reiterative and confusing, while time dynamics have not been analysed (i.e. differences among means of a single dietary treatment along the different samplings).

  • Discussion is shallow on the one hand, but on the other conclusions go too far.

            - Authors do not comment the lack of differences in microbiota diversity at OTU level, nor innate humoral factors.

            - L495-496 Authors talk about synergy between probiotic and vaccine, which has not been proven but only suggested by a significant survival. This hypothesis is at least described in a later paragraph (L548), but authors cannot talk about dendritic cell maturation and Th2 response in the light of their current results.

            - L497-498 Authors state such synergy develops humoral and non-specific immune responses, but they have no results demonstrating an improved non-specific immunity. L505-506 Where is a connection to innate immunity to be seen?

            - L505 The neutralizing potential of antibodies has not been studied here.      

  • Incongruences are spread throughout the manuscript.

Eg.       - L56 Is skin a physical barrier bacteria cross to enter the intestinal tract?

            - L57 Are lymphocytes considered a chemical barrier?

            - L59 How many kidneys do teleost have?

            - L197 Fish are anaesthetised after the sampling?

            - The used ELISA technique does not quantify IgM unless a standard is used for standard curve extrapolation. Besides, authors are detecting binding of specific IgM bound to S. agalactiae, not total IgM as they state (L178, L332, L536).

Reviewer 2 Report

In the manuscript entitled “Oral administration of probiotics (Bacillus subtilis and Lactobacillus plantarum) in Nile tilapia (Oreochromis niloticus) vaccinated and challenged with Streptococcus agalactiae”, the authors analyze the impact of a commercial probiotic formula known as AQUA PHOTO, on the potentialization of vaccination effect against S. agalactiae infection in the survival, immune response, and gut microbiota composition of Nile tilapia. They observed that the immunized fish exposed to the probiotic displayed an increased survival to S. agalactiae infection, even producing higher levels of IgM antibodies compared to the control group. Also, this vaccination and probiotic administration have an effect on Nile tilapia gut microbiota composition.

Even if the authors presented several interesting results, some aspects of their manuscript could be improved:

-       Please, provide the full composition of AQUA PHOTO commercial probiotic because the website is not available.

-       Since probiotics could have an impact on fish growth and development, did the authors evaluate the impact of this treatment on fish growth performance?

-       Could the authors clarify if the S. agalactiae strain has been deposited in any Bacterial strain collection? What is the exact strain name?

-       Why did the authors not include a control group non-exposed to the pathogenic bacterium? This information will provide the reader with a good background about the basal immunological level of the fish as well as the resilience of the microbiota of each challenged group at the end of the infection experiment. Please, clarify this point.

-       For immune response analysis, why do the authors focus on these three genes (MHCII, IgM and IL-1ß)? are they related specifically to S. agalactiae-induced infection immune response?

-       For microbiota composition analysis, mainly for taxonomical differences analysis, I strongly recommend using well-adapted statistical methods such as LefSE or ANCOM, instead of comparing only the most abundant taxonomic groups.

-       The abstract is quite redundant and should be shortened.

-       In the Discussion section, the authors hypothesize that the increased survival observed in the PRO group could be due to the immune response induction induced by the probiotics. However, what is known about anti-GBS activity by any of the bacterial species forming the AQUA PHOTO commercial probiotic? The antibacterial activity should not be ruled out.

-       The Discussion section is quite long and with some redundant information. I recommend shorting it to make it easier for the reader the comprehension of the main(s) message(s) of the manuscript. Idem for the Abstract section.

Some minor comments:

-       Line 49: please, include the scientific name of Nile tilapia.

-       Lines 67-68: there are more accurate definitions available in the literature for a probiotic such as Merrifield et al (doi: 10.1016/j.aquaculture.2010.02.007); “a probiotic organism can be regarded as a live, dead or component of a microbial cell, which is administered via the feed or to the rearing water, benefiting the host by improving disease resistance, health status, growth performance, feed utilization, stress response or general vigor, which is achieved at least in part via improving the host’s microbial balance or the microbial balance of the ambient environment.”

-       Lines 70-71: I am not convinced about this statement since there are many gram-positive pathogenic species (for fish and mammals). Please, develop or rewrite this sentence.

-       Table 1: if the probiotic mixture was homogenized in soybean oil, the inclusion of probiotic preparation in the diet has not effect in soybean oil amount of probiotic diet, why?

-       Line 134: “Four probiotic was used”. Is it correct? If yes, which ones?

-       Figure 1 quality should be improved and simplified since it is not easy to follow.

-       Lines 370-372: even if there are no differences in phagocytes and lysozyme activities, I strongly recommend showing these data (as Sup data).

-       Line 467: “Propionibacterium” instead of “Proprinebacterium”.

Reviewer 3 Report

- I suggest eliminating table 2, integrating the data in the text.

- Once the strain has been identified by PCR, I consider it unnecessary to carry out the sequencing.

- Although there are no differences in the levels of lysozymes and respiratory burst, the author should post the results.

- I believe that simultaneously expressing survival data and relative survival percentage (RPS) generates confusion. I believe that speaking only RPS would be better.